# Immunocytochemical Analysis of the Wall Ingrowths in the Digestive Gland Transfer Cells in *Aldrovanda vesiculosa* L. (Droseraceae)

**DOI:** 10.3390/cells11142218

**Published:** 2022-07-16

**Authors:** Bartosz J. Płachno, Małgorzata Kapusta, Piotr Stolarczyk, Piotr Świątek, Maciej Strzemski, Vitor F. O. Miranda

**Affiliations:** 1Department of Plant Cytology and Embryology, Institute of Botany, Faculty of Biology, Jagiellonian University in Kraków, 9 Gronostajowa St., 30-387 Cracow, Poland; 2Department of Plant Cytology and Embryology, Faculty of Biology, University of Gdańsk, 59 Wita Stwosza St., 80-308 Gdansk, Poland; malgorzata.kapusta@ug.edu.pl; 3Department of Botany, Physiology and Plant Protection, Faculty of Biotechnology and Horticulture, University of Agriculture in Kraków, 29 Listopada 54 Ave., 31-425 Cracow, Poland; piotr.stolarczyk@urk.edu.pl; 4Institute of Biology, Biotechnology and Environmental Protection, Faculty of Natural Sciences, University of Silesia in Katowice, 9 Bankowa St., 40-007 Katowice, Poland; piotr.swiatek@us.edu.pl; 5Department of Analytical Chemistry, Medical University of Lublin, Chodźki 4a, 20-093 Lublin, Poland; maciej.strzemski@poczta.onet.pl; 6Laboratory of Plant Systematics, School of Agricultural and Veterinarian Sciences, São Paulo State University (Unesp), Jaboticabal CEP 14884-900, Brazil; vitor.miranda@unesp.br

**Keywords:** arabinogalactan proteins, carnivorous plants, cell wall, Droseraceae, transfer cells, wall labyrinth, wall ingrowths, waterwheel plant

## Abstract

Carnivorous plants are unique due to their ability to attract small animals or protozoa, retain them in specialized traps, digest them, and absorb nutrients from the dissolved prey material; however, to this end, these plants need a special secretion-digestive system (glands). A common trait of the digestive glands of carnivorous plants is the presence of transfer cells. Using the aquatic carnivorous species *Aldrovanda vesiculosa*, we showed carnivorous plants as a model for studies of wall ingrowths/transfer cells. We addressed the following questions: Is the cell wall ingrowth composition the same between carnivorous plant glands and other plant system models? Is there a difference in the cell wall ingrowth composition between various types of gland cells (glandular versus endodermoid cells)? Fluorescence microscopy and immunogold electron microscopy were employed to localize carbohydrate epitopes associated with major cell wall polysaccharides and glycoproteins. The cell wall ingrowths were enriched with arabinogalactan proteins (AGPs) localized with the JIM8, JIM13, and JIM14 epitopes. Both methylesterified and de-esterified homogalacturonans (HGs) were absent or weakly present in the wall ingrowths in transfer cells (stalk cells and head cells of the gland). Both the cell walls and the cell wall ingrowths in the transfer cells were rich in hemicelluloses: xyloglucan (LM15) and galactoxyloglucan (LM25). There were differences in the composition between the cell wall ingrowths and the primary cell walls in *A. vesiculosa* secretory gland cells in the case of the absence or inaccessibility of pectins (JIM5, LM19, JIM7, LM5, LM6 epitopes); thus, the wall ingrowths are specific cell wall microdomains. Even in the same organ (gland), transfer cells may differ in the composition of the cell wall ingrowths (glandular versus endodermoid cells). We found both similarities and differences in the composition of the cell wall ingrowths between the *A. vesiculosa* transfer cells and transfer cells of other plant species.

## 1. Introduction

Carnivorous plants are able to attract small animals or protozoa, retain them in specialized traps, digest them, and absorb nutrients from the dissolved prey material. This allows them to cope with extreme habitats that are poor in macroelements [1,2,3]. Most carnivorous plants develop digestive glands, which produce and secrete lytic enzymes and are involved in the absorption of prey-derived nutrients. Therefore, digestive glands of carnivorous plants are specialized in the bidirectional transport of materials. Although these glands may vary in size, shape, and complexity, they consist of two main components: glandular and endodermoid cells in all genera of carnivorous plants [1]. Glandular cells are rich in organelles and perform various cellular activities. External glandular cells are exposed to stress associated with contact with bacteria or fungal pathogens and to mechanical damage caused by captured animals. Thus, they are an ideal model for studying plant cyto-architecture, membrane organization, and dynamics of organelle interactions [4,5,6,7,8,9]. In glandular cells, transport of materials into or out of the gland occurs through the symplast and apoplast. Adlassnig et al. [5] showed that many genera of carnivorous plants uptake nutrients not only via carriers but also by endocytosis. However, the endodermoid element of the gland (one or more cells with cuticularized or suberized cell walls) blocks transport via the apoplast and only the symplastic route of transport between glandular cells and the basal part of the gland is available [10,11,12].

A common characteristic of glandular cells of carnivorous plant digestive glands is the occurrence of cell wall ingrowths; hence, these cells are transfer cells [Juniper et al. 1989]. Glandular transfer cells have been detected in the digestive gland of species from nonrelated families of carnivorous plants e.g., in Droseraceae [13,14,15,16,17,18] and Lentibularaceae [19,20,21]. The transfer cells are characterized by the occurrence of secondary wall ingrowths, which are scaffolds for an amplified surface area of plasma membrane rich in nutrient transporters [22,23,24,25]. Transfer cells are an ancient and very important invention in land plant evolution. Plant fossils containing transfer cells were described from Devonian striata [26]. As reported by Offler et al. [25], a few main experimental systems to study the biology of transfer cells have been developed: cultured cotyledons of grain legumes [27,28,29], the basal endosperm of developing maize caryopses [30,31], root giant cells and syncytia [32,33,34], companion cells and phloem parenchyma transfer cells in leaf minor veins [35,36,37,38], and nucellar projection and aleurone/endosperm transfer cells of wheat and barley [39,40].

Despite the significant advances in understanding the molecular basis of the formation of wall ingrowths [25], the knowledge of the chemical composition (studied using immunogold labeling) of the wall labyrinth in transfer cells is still insufficient and based on studies in only a few angiosperm genera: *Pisum sativum*, *Vicia faba*, and *Elodea canadensis* [27,41,42]. Recently, nonflowering plants (bryophytes) have also been studied to determine the detailed chemical composition of the wall ingrowths in their placenta [43,44,45]. Therefore, there is a need to study the composition of wall ingrowths on various plant models.

Offler et al. [25] emphasized that one of the difficulties in studying transfer cells is their location, because they generally occur deep within the organ and are surrounded by other cell types. This restricts their accessibility for experimental investigation. The transfer cells in the digestive glands of many carnivorous plants are peripheral; therefore, access to these cells is much easier than in other experimental systems. The glandular transfer cells are involved in both secretion and absorption; thus, they can be studied at different phases of the digestive cycle: prey digestion and absorption of nutrients from animal remains. We proposed carnivorous plants as an excellent model for studies of wall ingrowths/transfer cells. In our previous paper [18], we showed the presence of arabinogalactan proteins (AGPs) in cell wall ingrowths in the digestive glands of *Dionaea muscipula* J.Ellis. The genus *Aldrovanda* (with one recent species *Aldrovanda vesiculosa* L., waterwheel plant) is a sister genus to *D. muscipula*, and both produce the snapping traps for catching small invertebrates [1,46,47,48,49,50]. We wanted to compare the *A. vesiculosa* digestive glands, which contain transfer cells, to those in *D. muscipula* in terms of the composition of wall ingrowths (in terms of the presence of arabinogalactans).

Here, we addressed the following questions: Is the cell wall ingrowth composition the same between carnivorous plant glands and other plant system models? Is there a difference in the cell wall ingrowth composition between various types of gland cells (glandular versus endodermoid)?

## 2. Materials and Methods

### 2.1. Plant Material

The *A. vesiculosa* L. plants (Polish clone) were collected from Mr. Maciej Kosiedowski’s (Tarnowskie Góry, Poland) private collection. For the digestive gland analysis, mature traps were taken from mature plants at the same stage of development.

### 2.2. Histological and Immunochemical Analysis

The traps were fixed overnight at 4 °C in 8% (*w*/*v*) paraformaldehyde (PFA, Sigma-Aldrich, Sigma-Aldrich Sp. z o.o. Poznan, Poland), 0.25% (*v*/*v*) glutaraldehyde (GA, Sigma-Aldrich, Sigma-Aldrich Sp. z o.o. Poznan, Poland) in PIPES buffer. PIPES buffer contains 50 mM PIPES (piperazine-N,N′-bis [2-ethanesulfonic acid], Sigma Aldrich, Sigma-Aldrich Sp. z o.o. Poznan, Poland), 10 mM EGTA (ethylene glycol-bis [β-aminoethyl ether] N,N,N′,N′-tetraacetic acid, Sigma Aldrich, Sigma-Aldrich Sp. z o.o. Poznan, Poland), and 1 mM MgCl_2_ (Sigma Aldrich, Sigma-Aldrich Sp. z o.o. Poznan, Poland), pH 6.8. Plant material was dehydrated through a graded series of ethanol (10-100%). For the analysis of the localization of the major cell wall polysaccharides and glycoproteins, specimens were infiltrated with raising changes of LR White Resin (Polysciences Europe GmbH, Hirschberg an der Bergstrasse, Germany) mixed with 100% ethanol, up to pure resin (at 4 °C, each change for 2 h and in second change of pure resin–overnight). Then placed in gelatin capsules in fresh pure LR White resin and polymerized at 55 °C. Procedure was repeated twice, and plant material was sectioned using a microtome (an RMC Power XT ultramicrotome RMC Boeckeler or a Leica ultracut UCT ultramicrotome). The rehydrated sections were blocked with 1% bovine serum albumin (BSA, Sigma-Aldrich, Sigma-Aldrich Sp. z o.o. Poznan, Poland) in PBS buffer and incubated with the following primary antibodies (purchased from Plant Probes, Leeds, UK)—anti-AGP: JIM8, JIM13; JIM14 [51,52,53,54], antipectin: JIM5, JIM7, LM19, LM5, LM6 [51,55,56,57]; antihemicelluloses: LM25, LM15 [56,57,58], and antiheteromannan [59] overnight at 4 °C. All the primary antibodies were used in a 1:20 dilution. They were purchased from Plant Probes, UK, and the goat antirat secondary antibody conjugated with FITC was purchased from Abcam (Abcam plc, Cambridge, UK). The chromatin in the nuclei was stained with 7 µg/mL DAPI (Sigma Aldrich, Sigma-Aldrich Sp. z o.o. Poznan, Poland) diluted in PBS buffer and the samples were then cover-slipped using a Mowiol mounting medium: a mixture of Mowiol^®^ 4-88 (Sigma Aldrich, Sigma-Aldrich Sp. z o.o. Poznan, Poland) and glycerol for fluorescence microscopy (Merck, Merck Sp. z o. o., Warsaw, Poland) with the addition of 2.5% DABCO (The Carl Roth GmbH + Co. KG, Karlsruhe, Germany). They were viewed using a Nikon Eclipse E800 microscope (Precoptic, Warsaw, Poland) or a Leica DM6000B microscope (KAWA.SKA Sp. z o.o., Piaseczno, Poland). Photos were acquired as Z stacks and deconvolved using 5 iterations of a 3D nonblind algorithm (AutoQuant ™, Media Cybernetics Inc., Rockville, MD, USA). In order to maximize the spatial resolution, the images are presented as maximum projections. The stacks were obtained using a Leica DM6000B microscope equipped with a GFP filter. At least two different replications were performed for each of the analyzed traps, and about 5 to 10 sections from each organ were analyzed for each antibody used. Negative controls were created by omitting the primary antibody step, which caused no fluorescence signal in any of the control frames for any of the stained slides (Appendix A).

Mean values of fluorescence intensity were calculated from the GFP channel using the LAS AF Quantify module (Leica Microsystems, Wetzlar, Germany). The regions of interest were selected manually for distinctive cell walls of 3 glands from 3 different traps (n = 3). The data were analyzed statistically using Statistica 13 (StatSoft Polska Sp. z o.o., Cracow, Poland). For comparisons of the mean values, an analysis of variance (one-way ANOVA) followed by post hoc Tukey’s honestly significant difference test was used. For all analyses, the significance level was estimated at *p*  <  0.05 (Appendix A). Boxplots were created using Statistica 13 (StatSoft Polska Sp. z o.o., Cracow, Poland).

Semithin sections (0.9–1.0 μm thick) were prepared for light microscopy and stained for general histology using aqueous methylene blue/azure II (MB/AII) for 1–2 min. A histochemical procedure with fixed material using the PAS reaction (periodic acid-Schiff reaction) was performed to detect the polysaccharides (wall ingrowths) [60]. Calcofluor white staining was used to detect cellulose in the cell wall ingrowths [60].

### 2.3. Immunogold Labeling Distribution of AGP, HG, Hemicellulose, and Mannan

A Leica Ultracut UCT ultramicrotome was used to prepare ultrathin sections (50 nm). They were blocked in 1% BSA (Aurion, Wageningen, the Netherlands) in PBS buffer for 15 min and then incubated in primary antibodies in a 1:10 dilution overnight at 4 °C. Followed by washes in PBS buffer (6 × 5 min) and incubated with the goat antirat secondary antibody conjugated with 10 nm colloidal gold (Sigma Aldrich, Poland) in a 1:50 dilution for 2 h, followed by washing in PBS buffer and distilled water. Negative controls were created by omitting the primary antibody step (Appendix A). Lead citrate (Microshop, PIK Instruments Sp. z o.o., Piaseczno, Poland) and URANYLess (Microshop, PIK Instruments Sp. z o.o., Piaseczno, Poland) were added as contrasting agents. The cells were visualized using a Jeol JEM 100 SX microscope (JEOL, Tokyo, Japan) at 80 kV in the Department of Cell Biology and Imaging, Institute of Zoology, Jagiellonian University in Kraków or a Hitachi UHR FE-SEM SU 8010 microscope at 25 kV, housed at the University of Silesia in Katowice.

## 3. Results

### 3.1. General Gland Structure and Histochemistry

Each digestive gland consisted of two basal (endodermoid) cells, four stalk cells, and a head with about 12 secretory cells (Figure 1A). All these cells were transfer cells (Figure 1B,C). The PAS reaction demonstrated that the wall ingrowths contained carboxylated polysaccharides (Figure 1D). The calcofluor white staining demonstrated the presence of cellulose in the cell wall ingrowths (Figure 1E). The cell wall ingrowths represented the reticulate type (Figure 1B,C), according to the classification proposed by Talbot et al. [30]. The presence or absence of each examined epitope detected in the digestive glands are shown in Table 1.

### 3.2. AGP Distribution

A strong fluorescence signal of AGP epitope recognized by JIM8 was observed in the walls of the digestive gland cells (Figure 2A,B). An intense signal of this epitope was recorded in the wall ingrowths. The immunogold labeling with JIM8 showed that the pectic AGP epitopes were localized in the walls and wall ingrowths of the gland cells (Figure 2C). The AGP epitope recognized by JIM13 was present in the wall ingrowths of the digestive gland cells (intense signal) (Figure 2D,E). A diffuse signal of this epitope was observed in the cytoplasm of the digestive gland cells. The immunogold labeling with JIM13 showed that the pectic AGP epitopes were localized in the walls and wall ingrowths of the gland cells (Figure 2F–I). The gold particles also appeared in the cytoplasm and vesicles of the gland cells. The epitope recognized by JIM14 was restricted to the gland cell walls, mainly to the cell wall ingrowths (intense signal) (Figure 2J,K). The immunogold labeling with JIM14 showed that the AGP epitopes were localized in cell walls and in the wall ingrowths in the gland cells (Figure 2L).

### 3.3. Homogalacturonan Distribution

A strong fluorescence signal detected by JIM5 (low methylesterified HG) was observed in the walls of ordinary epidermal and parenchyma cells of the traps. In the digestive gland, this epitope was present in the walls of basal cells adjacent to the epidermal and parenchyma cells. In the basal cells, there was no signal showing this epitope in the external (endodermoid) cell walls and in the walls between the basal cells (Figure 3A,B). Additionally, there was no signal of this epitope in the walls of the lower part of the stalk cells. In the head cells, the fluorescence signal detected by JIM5 was located in the cell walls. In some glands, only a weak signal was visible as dots in the head cell walls (Figure 3B). No signal was observed in the wall ingrowths (Figure 3A,B); also, no gold particles were observed (Figure 3C). The fluorescence signal detected by LM19 (low methylesterified HG) was observed in the walls of epidermal and parenchyma cells of the traps and in the gland cell walls. This epitope occurred in the middle lamella in the head cell walls. No signal was observed in the wall ingrowths (Figure 3D,E). The immunogold labeling with LM19 showed that the pectic HG epitopes were localized in the glandular cell walls (Figure 3F), and a very low amount of gold particles was observed in the wall ingrowths (Figure 3F). An intense fluorescence signal from highly esterified HG (detected by JIM7) was observed in the walls of the epidermal and parenchyma cells of the traps (Figure 3G,H). This epitope occurred in the basal cell walls and also in the wall labyrinth in these cells. This epitope was also detected in the basal part of the stalk cell walls. A less intensive signal of this epitope occurred in the head cell walls. No signal was observed in the wall ingrowths of both stalk cells and head cells. The immunogold labeling with JIM7 showed that the pectic HG epitopes were localized in the walls of the ordinary epidermal and parenchyma cells (Figure 4A). An intense signal from the pectic polysaccharide (1–4)-β-D-galactan (detected by LM5) was observed in the walls of epidermal and parenchyma cells of the traps and in the cell walls of sensory trichome cells (Figure 4B). This epitope occurred in the gland cell walls; however, no signal was observed in the wall ingrowths (Figure 4C,D). The immunogold labeling with LM5 showed that the pectic polysaccharide (galactan) epitopes were localized in the cell walls (Figure 4E), but not in the wall ingrowths. The signal from the pectic polysaccharide alpha-1,5-arabinan (detected by LM6) (Figure 4F,G) was observed in the walls of epidermal and parenchyma cells of the traps. This epitope occurred in the wall labyrinth in the basal cells but was absent (no signal) in the wall ingrowths in stalk and head cells (Figure 4G). No gold particle was observed in the wall ingrowths (Figure 4H).

### 3.4. Hemicellulose and Heteromannan Distribution

A signal from xyloglucan (detected by LM15) was observed in the external walls of the trap epidermal cells. This epitope occurred in the cell walls of all digestive gland cell types. An intense signal of this epitope occurred in the wall ingrowths (Figure 5A,B). In the head cells, a less intense signal occurred in the corners of the middle lamella and in the corners between cells. The immunogold labeling with LM15 showed that the xyloglucan epitopes were localized in the wall ingrowths in the gland cells (Figure 5C–E). Especially dense labeling of xyloglucan was observed in the ingrowths of the basal cell wall (Figure 5D). A signal of xyloglucan (labeled with LM25) was observed in the cell walls in the trap wall (parenchyma, epidermis) and digestive glands (Figure 5F,G). This epitope occurred in the walls of all types of digestive gland cells. An intense signal from xyloglucan occurred in the wall ingrowths. The immunogold labeling with LM25 showed that the xyloglucan epitopes were localized in the walls and wall ingrowths in the gland cells (Figure 5H,I). No signal from heteromannan (detected by LM21 and LM22) was detected in the cell wall ingrowths (Figure 6A,B).

### 3.5. Statistical Analysis

One-way ANOVA analyses were performed for four distinctive cell walls with ingrowths: head cell wall, ordinary epidermal cell wall, stalk cell wall, and basal cell wall. All analyses showed statistical significance (Appendix A).

#### 3.5.1. Head Cell Wall

The strongest signals were observed for AGP epitopes localized in head wall cells (Figure 7) recognized by JIM8, JIM13, and JIM14. The signals for all three investigated AGPs were also significantly higher in comparison with those for pectins (JIM5, JIM7, and LM6 epitopes). The signals for pectin recognized by LM6 and LM19 were significantly lower than the AGPs detected by JIM13 and JIM14. The lowest signals were observed for pectins (JIM5, JIM7, and LM6 epitopes) and both heteromannans recognized by LM21 and LM22.

#### 3.5.2. Ordinary Epidermal Cell Wall

The strongest signals in the ordinary epidermal cell wall (Figure 8) were detected for AGP recognized by JIM13, pectin (LM5 epitope), and hemicelluloses (LM 15 and LM25 epitopes). The epitopes recognized by JIM13, LM5, and LM25 were significantly stronger than in the control (CRTL) and both selected heteromannans (LM21 and LM22 epitopes). The signal for hemicellulose recognized by LM25 was significantly higher than that for the control (CTRL), two AGPs (JIM8 and JIM14 epitopes), and both heteromannans (LM21 and LM22 epitopes). The signal for AGP recognized by JIM8 was significantly lower than for one of the pectins (LM5 epitope) and one of the hemicelluloses (LM25 epitope). The signal for the JIM14 epitope was significantly lower only than that for the other AGP (JIM13 epitope), one of the pectins (LM5 epitope), and one of the hemicelluloses (LM25 epitope). Additionally, another AGP detection (JIM13 epitope) and the signal for pectin recognized by LM5 were significantly stronger than the control (CTRL) and both AGPs detected with JIM8 and JIM14. Signals detected by JIM7 and LM6 did not show any significance to any other epitopes analyzed and control (CTRL). The signals for the epitopes recognized by LM19 (pectin epitope) and LM15 (hemicellulose epitope) were stronger only in comparison to the control (CTRL). The lowest signal was detected for one of the heteromannans (LM21 antibody).

#### 3.5.3. Stalk Cell Wall

The strongest signals in the stalk cell wall (Figure 9) were observed for both hemicelluloses (used LM15 and LM25 antibodies). Hemicellulose recognized by LM25 showed significantly stronger signal than the control (CTRL), all pectins (JIM5, JIM7, LM5, LM6 and LM19 epitopes), and both heteromannans (LM21 and 22 epitopes). Two pectins recognized by JIM7 and LM19, also both heteromannans (LM21 and 22 epitopes), did not show any differences with the control (CTRL). Pectins detected with JIM5, JIM7, and JIM19 antibodies showed significantly lower than two AGPs (JIM13 and JIM14 epitopes) and both hemicelluloses (LM15 and LM25 epitopes). The signal for JIM8 epitope was significantly higher than the control (CTRL) and both heteromannans (used LM21 and LM22antibodies) but lower than one of the hemicelluloses detected with LM15 antibody. The signals for both AGPs detected with JIM13 and JIM14 antibodies were significantly higher than control (CTRL), all analyzed pectins (JIM5, JIM7, LM6 and LM19 epitopes) and both heteromannans (LM21 and 22 epitopes). The lowest signals were observed for both heteromannans (used LM21 and LM22 antibodies), without difference from the control material (CTRL).

#### 3.5.4. Ingrowths of Basal Cell Wall

The significantly strongest signals for the basal cell wall ingrowths (Figure 10) were observed for both hemicelluloses (LM15 and LM25 epitopes) and two AGPs recognized by JIM13 and JIM14. The signal detected by LM15 was stronger than that for the control (CTRL) and any other analyzed epitopes. The signals detected for pectins recognized by JIM5, JIM7, and LM19 and both heteromannans (LM21 and LM22 epitopes) did not show any differences compared to the control (CTRL). Both heteromannan signals (LM21 and LM22 epitopes) were significantly lower than the signal for AGP (JIM13 and JIM14 epitopes), one pectin (LM6 epitope), and both hemicelluloses (detected by LM15 and LM25). Two AGP signals detected by the JIM13 and JIM14 epitopes were lower only than those for both hemicelluloses (LM15 and LM25 epitopes) but significantly stronger than in the control (CTRL) and any other analyzed epitopes.

## 4. Discussion

We showed that the cell wall ingrowths in the transfer cells of *A. vesiculosa* glands were rich in AGPs. The wall ingrowths were similar in their AGP composition (JIM8, JIM13, JIM14) to the wall ingrowths in the transfer cells of *D. muscipula* digestive glands [18]. It was shown that the epitope recognized by JIM14 was a useful marker of the digestive glands of *D. muscipula*. Here, also, the occurrence of AGPs (recognized by JIM14) was limited to the glands. This suggests that these AGPs may play an important role in the gland function. AGPs were reported from the wall ingrowths in transfer cells of both angiosperms [27,41,42,61] and lower plants [43,44,45]. Vaughn et al. [27] experimentally proved that AGPs play a role in the development of cell wall ingrowths. The wall ingrowth density was reduced when developing transfer cells were exposed to β-D-glucosyl Yariv reagent. McCurdy et al. [62] proposed that, since AGPs have a role in coordinating the required localized assembly of wall components, they play a role in the formation of cell wall ingrowths. Actin filaments anchored to arabinogalactan proteins are crucial for the formation of cell wall ingrowths [25]. In both *D. muscipula* and *A. vesiculosa*, the cell wall ingrowths in the gland cells were rich in AGPs in mature traps. It is known that AGPs play various roles in plants: developmental processes, calcium capacitor, signaling, response to biotic and abiotic stress, and sexual reproduction [18,63,64,65,66,67,68]. In *Dionaea muscipula*, we found differences in the occurrence of AGPs (labeled with JIM8 and JIM13) in the walls of gland secretory cells between unfed and fed traps. Thus, it seems that AGPs play a role in the digestion–absorption cycle. AGPs were also localized in secretory cells of *Drosera capensis* tentacles. [69]. As reported by Lichtscheidl et al. [9], in these cells, membrane recycling to prevacuolar compartments and vacuoles occurs for arabinogalactan proteins. This may be connected with prey digestion and subsequent endocytosis of nutrients. Here, in the cells of the *A. vesiculosa* glands, we also detected AGPs in the cytoplasm and vesicles. Schulze et al. [70] and Bemm et al. [71] proposed that *Dionaea* uses stress pathway-related processes to form an active digestive system. Hence, Płachno et al. [18] suggested that this could explain the accumulation of AGPs in the digestive glands, especially given the evidence for AGP overproduction under stress [64]. A similar mechanism may operate in the *A. vesiculosa* digestive glands and in other carnivorous plant species; however, more research is needed.

The methylesterified and demethylesterified HG epitopes were found abundantly in the cell walls of epidermal and parenchyma cells of the traps. In turn, in the case of HGs, the cell walls of the glands were not similar to the cell walls of the underlying epidermis and parenchyma cells. The stalk and head cell walls were poor in HG epitopes recognized by JIM5 and JIM7. The epitope recognized by JIM7 occurred in the basal parts of the stalk cells. This may be related to the mechanical role of these parts of the cells, which have to support the expanded parts of the stalk and head cells. Similarly, HG epitopes recognized by JIM7 in *Arabidopsis thaliana* were present exclusively in the inner wall layer at the trichome base [72]. Methylesterified HGs may enhance cell wall strength [73], which may explain the occurrence of these HGs in the basal parts of trichomes and glands. In contrast, in *Solanum* glandular trichomes, HGs recognized by JIM7 occurred in head cell walls [74]. Bowling et al. [75] showed distinct pectin domains in *Galium* trichomes, which facilitate the formation of highly curved trichomes. The authors suggested that highly de-esterified pectins may be involved in the interactions between the cell wall and the surface waxes. In *A. vesiculosa*, the low methylesterified HGs (recognized by LM19) occurred in the gland cells. During development of cannabis glandular trichomes, both methylesterified and demethylesterified HG epitopes were found in the walls of glandular cells, but later the amount of these epitopes decreased significantly after the separation of the subcuticular wall [76]. As suggested by Bergau et al. [74], pectin demethylation seems to play a role in the lysis of the inner cell wall and formation of the intercellular cavity in the head of *Solanum* glandular trichomes.

Due to the positive PAS reaction, we expected HGs to be present in the cell wall ingrowths in the *A. vesiculosa* gland cells, since methylesterified HGs were found in the wall ingrowths in other plant species [27,44,45]. However, we did not detect HGs recognized by JIM5, JIM7, LM19, LM5, and LM6 (using light microscopy) in the cell wall ingrowths of the stalk and head cells. The cell wall ingrowths in the basal cells had a different composition of HGs than the cell wall ingrowths of the stalk and head cells, i.e., we detected HGs recognized by JIM7 and LM6 in the wall ingrowths of the basal cells. Probably such differences in the wall ingrowths between the basal cells and the stalk and head cells may be related to the different functions of these cells. Basal cells mainly have transport functions, while head and stalk cells additionally have a secretory character. Noteworthy, the LM6 antibody not only recognizes arabinan, rhamnogalacturonan-I/(1–5)-α-L-arabinan but also labels AGPs; thus, the differences between the wall ingrowths in the basal cells and those in the stalk and head cells may be associated with the occurrence of AGP. Strong labeling by LM6 was observed in the cell walls of companion cells of minor veins in the lamina of chicory, which are transfer cells, by Sun et al. [77]. Similarly, this epitope occurred in the wall ingrowths of transfer cells in mature cotyledons of broad bean [27].

Both methylesterified and de-esterified HGs pectins were absent from the wall ingrowths in transfer cells of *Elodea canadensis* leaves [42], but occurred in the cell wall ingrowths in transfer cells of *Marchantia polymorpha* [43], *Phaeoceros carolinianus*, *P. laevis* [45], and *Physcomitrium patens* [44]. HG recognized by JIM7 occurred in the cell wall ingrowths in transfer cells of *Vicia faba* [27] and *Pisum sativum* [41]. Henry and Renzaglia [44] suggested that, due to their roles in cell wall properties and mechanics, HG may facilitate nutrient uptake by membrane transport proteins, which is important in transfer cell functioning.

We did not find galactan in the cell wall ingrowths of *A. vesiculosa* gland cells. Similarly, Ligrone et al. [42] did not detect this compound in the wall ingrowths in transfer cells of *E. canadensis* leaves. However, galactan was found in the wall ingrowths in transfer cells of *V. faba* [27] and bryophytes [44,45]. In contrast, Vaughn et al. [27] did not observe galactan in the outer region of wall ingrowths. This proves the differences in the composition of the cell walls in different species, which may be related to different physiology or evolutionary affinities.

We showed that the cell walls and cell wall ingrowths in the transfer cells of *A. vesiculosa* glands were rich in hemicelluloses: xyloglucan (LM15) and galactoxyloglucan (LM25). These xyloglucans were also recorded in the cell wall ingrowths in bryophytes [43,44,45]. Xylan/arabinoxylan (recognized by LM11) was recorded in the fibrillar core of wall ingrowths in *Elodea* [42]. Xyloglucans play a key role in the loosening and tightening of cellulose microfibrils, which enables the cell to change its shape during growth, differentiation, and enlargement [78,79]. Thus, the occurrence of xyloglucan in the *A. vesiculosa* glands seems very important for gland functioning. In this species, Muravnik [14] observed enlargement of secretory gland cells after feeding. Given the cytological changes occurring in the glands during prey digestion (digestive enzyme secretion) and nutrient absorption [1,7,8], it can be expected that hemicelluloses are present in the glandular structures of other species of carnivorous plants.

## 5. Conclusions

Because we found that even in the same organ/structure (gland), transfer cells may differ in the composition of the cell wall ingrowths (glandular endodermoid cells), further research should be more comprehensive and cover different types of glandular structures and different types of tissue in the analyzed species.

## Figures and Tables

**Figure 1 cells-11-02218-f001:**
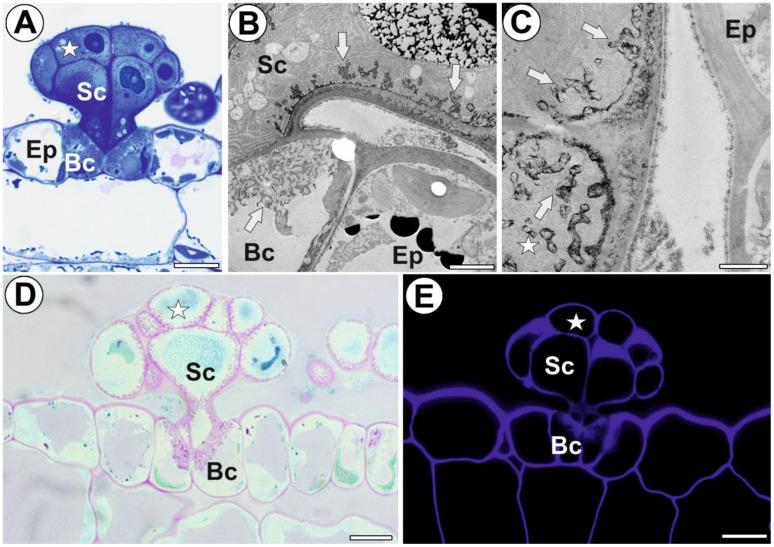
Structure of the digestive gland of *A.vesiculosa*. (**A**) A semithin section of the digestive gland; secretory cell (star), stalk cell (Sc), basal cell (Bc), ordinary epidermal cell (Ep), bar 10 µm. (**B**,**C**) Ultrastructure of digestive glands, localization of cell wall ingrowths (white arrow); head cell (star), stalk cell (Sc), basal cell (Bc), ordinary epidermal cell (Ep), bar 1 µm. (**D**) Digestive gland, PAS reaction; head cell (star), stalk cell (Sc), basal cell (Bc), bar 10 µm. (**E**) Digestive gland, calcofluor white staining; head cell (star), stalk cell (Sc), basal cell (Bc), bar 10 µm.

**Figure 2 cells-11-02218-f002:**
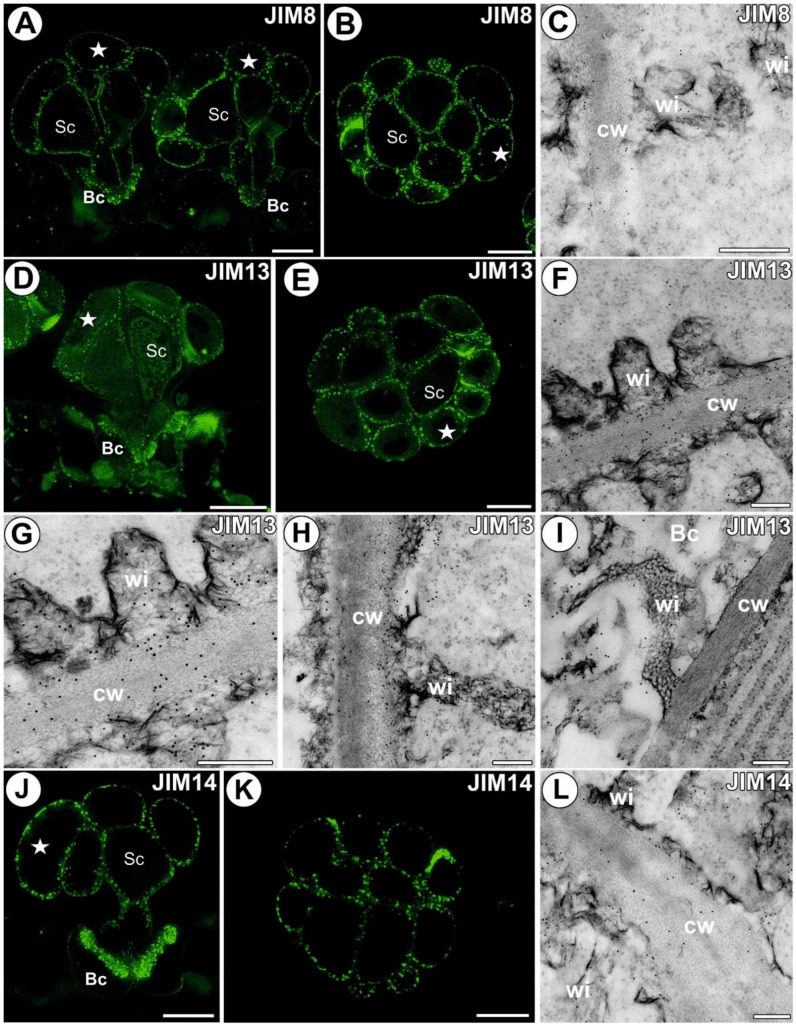
Arabinogalactan proteins detected in the *A. vesiculosa* trap. (**A**) Arabinogalactan proteins (labeled with JIM8) detected in the gland; secretory cell (star), stalk cell (Sc), basal cell (Bc), bar 10 µm. (**B**) Arabinogalactan proteins (labeled with JIM8) detected in the gland, paradermal section; secretory cell (star), stalk cell (Sc), bar 10 µm. (**C**) Immunogold labeling of wall ingrowths with JIM8 in the gland cell; wall ingrowths (wi), cell wall (cw); bar 400 nm. (**D**) Arabinogalactan proteins (labeled with JIM13) detected in the gland; secretory cell (star), stalk cell (Sc), basal cell (Bc), bar 10 µm. (**E**) Arabinogalactan proteins (labeled with JIM13) detected in the gland, paradermal section; secretory cell (star), stalk cell (Sc), bar 10 µm. (**F**–**H**) Immunogold labeling of wall ingrowths with JIM13 in the glandular cells; wall ingrowths (wi), cell wall (cw), bar 200 nm. (**I**) Immunogold labeling of wall ingrowths with JIM13 in the basal cell (Bc); wall ingrowths (wi), cell wall (cw); bar 200 nm. (**J**) Arabinogalactan proteins (labeled with JIM14) detected in the gland; secretory cell (star), stalk cell (Sc), basal cell (Bc), bar 10 µm. (**K**) Arabinogalactan proteins (labeled with JIM14) detected in the gland, transverse section; bar 10 µm. (**L**) Immunogold labeling of wall ingrowths with JIM14 in the gland cell; wall ingrowths (wi), cell wall (cw); bar 200 nm.

**Figure 3 cells-11-02218-f003:**
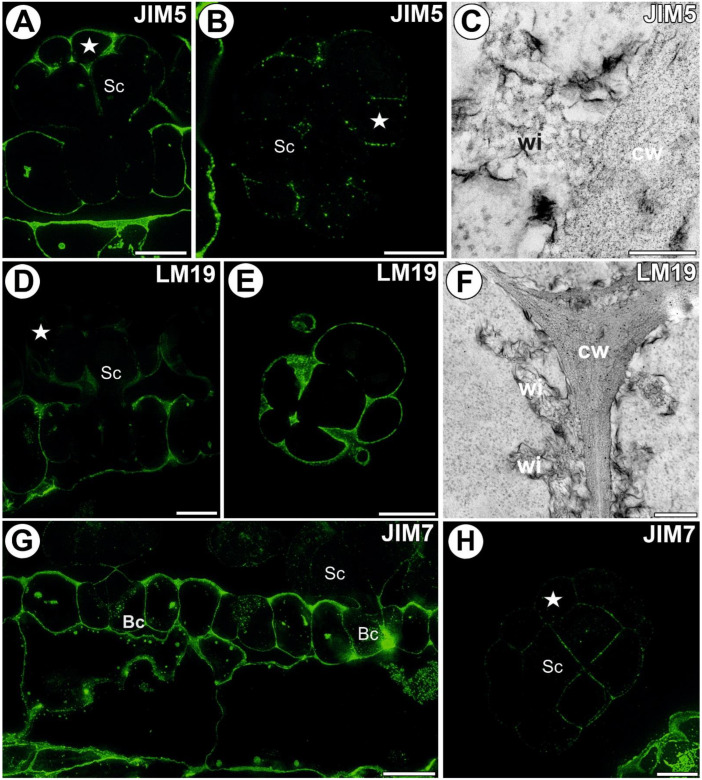
Homogalacturonans detected in the *A. vesiculosa* trap. (**A**) HG (labeled with JIM5) detected in the gland; secretory cell (star), stalk cell (Sc), basal cell (Bc), bar 10 µm. (**B**) HG (labeled with JIM5) detected in the gland, transverse section; secretory cell (star), stalk cell (Sc), bar 10 µm. (**C**) HG (labeled with JIM5) detected in the gland, wall ingrowths (wi), cell wall (cw), bar 200 nm (**D**) HG (labeled with JIM19) detected in the gland; secretory cell (star), stalk cell (Sc), basal cell (Bc), bar 10 µm. (**E**) HG (labeled with JIM19) detected in the gland, transverse section; bar 10 µm. (**F**) Immunogold labeling of wall ingrowths with JIM19 in the glandular cells; wall ingrowths (wi), cell wall (cw); bar 300 nm. (**G**) HG (labeled with JIM7) detected in the glands and trap wall; secretory cell (star), stalk cell (Sc), basal cell (Bc), bar 10 µm. (**H**) HG (labeled with JIM7) detected in the gland, paradermal section; secretory cell (star), stalk cell (Sc), bar 10 µm.

**Figure 4 cells-11-02218-f004:**
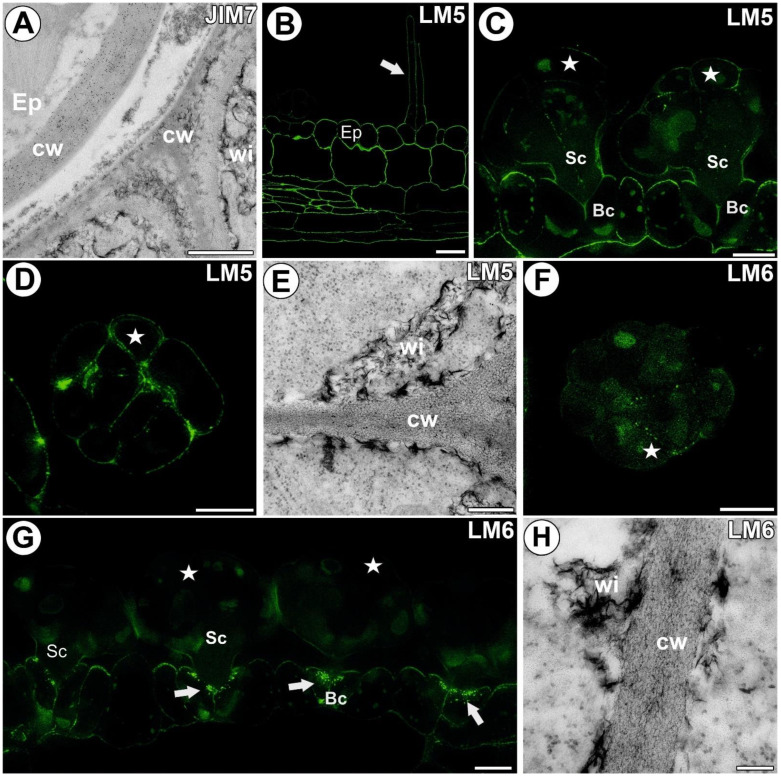
Homogalacturonans detected in the *A. vesiculosa* trap. (**A**) Immunogold labeling of the cell wall with JIM19 in the ordinary epidermal (Ep) and glandular cells; wall ingrowths (wi), cell wall (cw); bar 900 nm. (**B**) HG (labeled with LM5) detected in the trap wall; epidermal cell (Ep), sensory trichome (white arrow), bar 10 µm. (**C**) HG (labeled with LM5) detected in the glands; secretory cell (star), stalk cell (Sc), basal cell (Bc), bar 10 µm. (**D**) HG (labeled with LM5) detected in the gland, paradermal section; secretory cell (star), bar 10 µm. (**E**) Immunogold labeling of the wall ingrowths with LM5 in the glandular cells; wall ingrowths (wi), cell wall (cw); bar 300 nm. (**F**) HG (labeled with LM6) detected in the gland, transverse section; secretory cell (star), bar 10 µm. (**G**) HG (labeled with LM6) detected in the glands and trap wall; secretory cell (star), stalk cell (Sc), basal cell (Bc), wall labyrinth in the basal cell (white arrow), bar 10 µm. (**H**) Immunogold labeling of the wall ingrowths with LM6 in the glandular cells; wall ingrowths (wi), cell wall (cw); bar 100 nm.

**Figure 5 cells-11-02218-f005:**
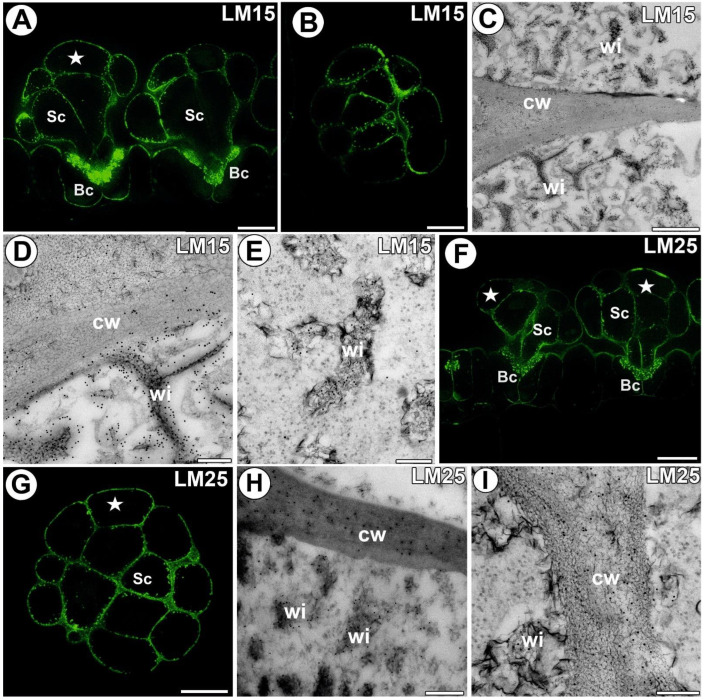
Xyloglucan detected in the *A. vesiculosa* trap. (**A**) Xyloglucan (labeled with LM15) detected in the glands; secretory cell (star), stalk cell (Sc), basal cell (Bc), bar 10 µm. (**B**) Xyloglucan (labeled with LM15) detected in the gland, transverse section; bar 10 µm. (**C**,**D**) Immunogold labeling of wall ingrowths with LM15 in the basal cells; wall ingrowths (wi), cell wall (cw); bar 800 and 200 nm, respectively. (**E**) Immunogold labeling of wall ingrowths with LM15 in the glandular cells; wall ingrowths (wi), cell wall (cw); bar 200 nm. (**F**) Xyloglucan (labeled with LM25) detected in the glands and trap wall; secretory cell (star), stalk cell (Sc), basal cell (Bc), bar 10 µm (**G**) Xyloglucan (labeled with LM25) detected in the gland, transverse section; secretory cell (star), stalk cell (Sc), bar 10 µm. (**H**,**I**) Immunogold labeling of wall ingrowths with LM15 in the gland cells; wall ingrowths (wi), cell wall (cw); bar 300 nm and bar 200 nm.

**Figure 6 cells-11-02218-f006:**
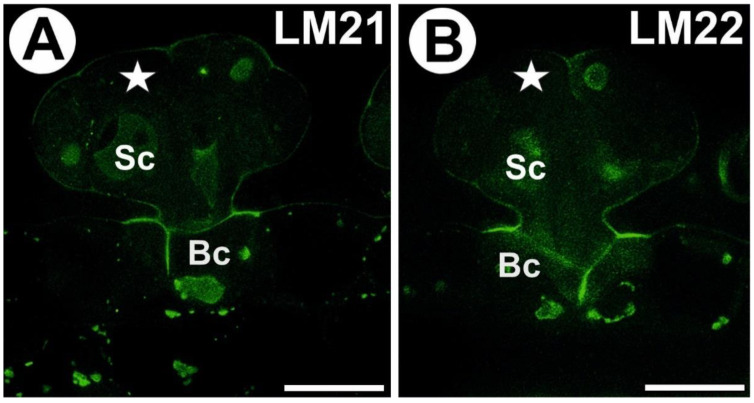
Heteromannan labeling in the *Aldrovanda vesiculosa* trap. (**A**) Heteromannan (labeled with LM21) detection in the gland, no signal in cell wall ingrowths; secretory cell (star), stalk cell (Sc), basal cell (Bc), bar 10 µm. (**B**) Heteromannan (labeled with LM22) detection in the gland, no signal in cell wall ingrowths; secretory cell (star), stalk cell (Sc), basal cell (Bc), bar 10 µm.

**Figure 7 cells-11-02218-f007:**
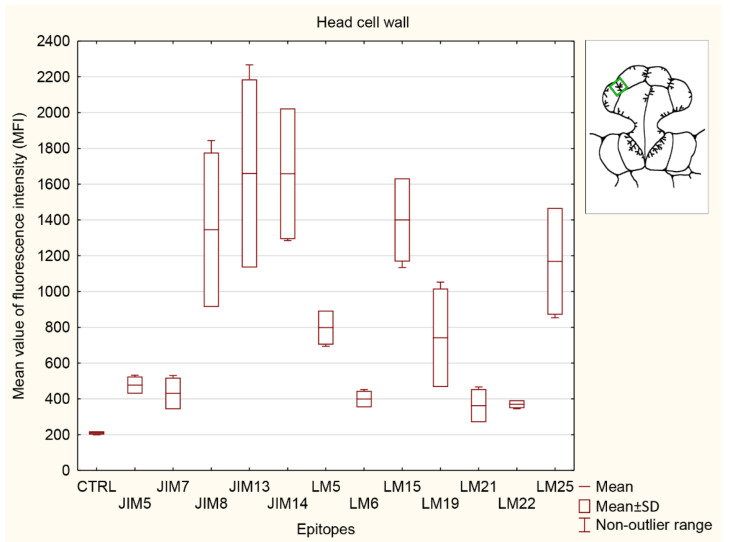
Quantification of immunofluorescence labeling for the head cell wall. Mean values of AGP fluorescence intensity (MFI) for negative control reaction (CTRL) and for labeled AGPs (JIM14, JIM8, and JIM13 epitopes), pectins (JIM5, JIM7, LM5, LM6, and LM19 epitopes), hemicelluloses (LM 15 and LM25 epitopes), and heteromannans (LM21 and LM22 epitopes) in 3 glands from 3 different traps (n = 3).

**Figure 8 cells-11-02218-f008:**
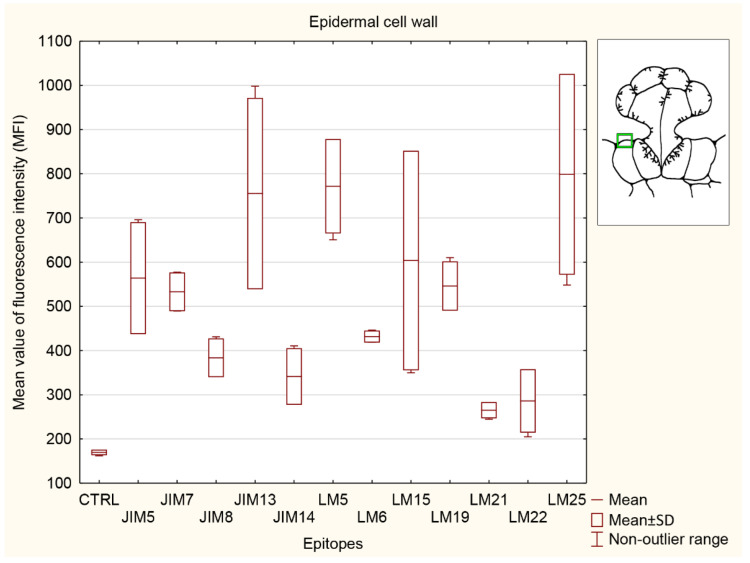
Quantification of immunofluorescence labeling for the ordinary epidermal cell wall. Mean values of AGP fluorescence intensity (MFI) for negative control reaction (CTRL) and for labeled AGPs (JIM14, JIM8, and JIM13 epitopes), pectins (JIM5, JIM7, LM5, LM6, and LM19 epitopes), hemicelluloses (LM 15 and LM25 epitopes), and heteromannans (LM21 and LM22 epitopes) in 3 glands from 3 different traps (n = 3).

**Figure 9 cells-11-02218-f009:**
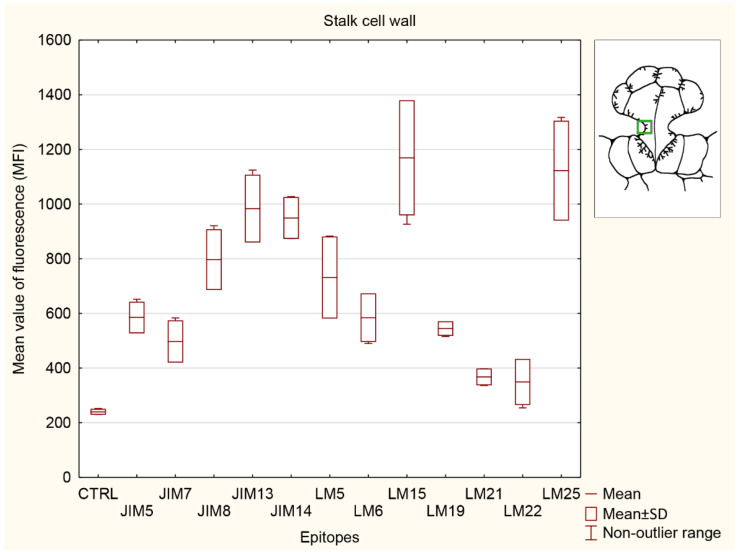
Quantification of immunofluorescence labeling for the stalk cell wall. Mean values of AGP fluorescence intensity (MFI) for negative control reaction (CTRL) and for labeled AGPs (JIM14, JIM8, and JIM13 epitopes), pectins (JIM5, JIM7, LM5, LM6, and LM19 epitopes), hemicelluloses (LM15 and LM25 epitopes), and heteromannans (LM21 and LM22 epitopes) in 3 glands from 3 different traps (n = 3).

**Figure 10 cells-11-02218-f010:**
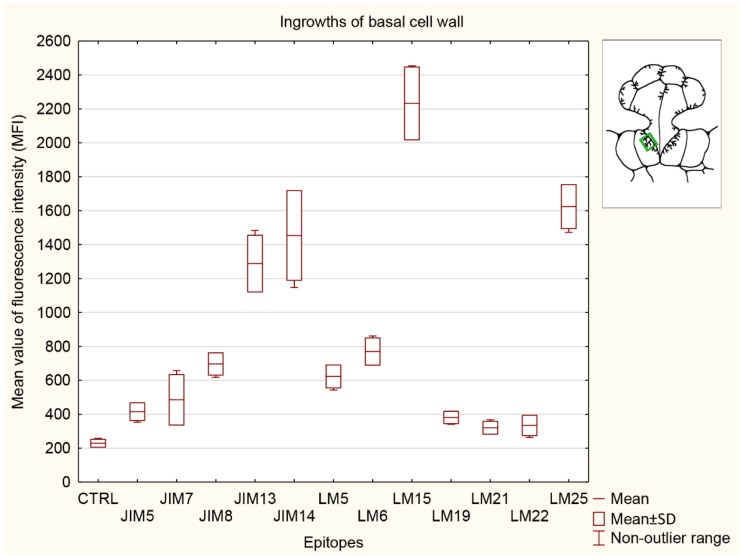
Quantification of immunofluorescence labeling for ingrowths of basal cell wall. Mean values of AGP fluorescence intensity (MFI) for negative control reaction (CTRL) and for labeled AGPs (JIM14, JIM8, and JIM13 epitopes), pectins (JIM5, JIM7, LM5, LM6, and LM19 epitopes), hemicelluloses (LM 15 and LM25 epitopes), and heteromannans (LM21 and LM22 epitopes) in 3 glands from 3 different traps (n = 3).

**Table 1 cells-11-02218-t001:** Summary of antibody labeling of selected cell walls in *Aldrovanda vesiculosa* digestive gland.

Epitope/ Type of Cell Wall/Cell	Arabinogalactans			Homogalacturonans					Hemicelluloses		Heteromannans	
	JIM8	JIM13	JIM14	JIM5	JIM7	LM5	LM6	LM19	LM15	LM25	LM21	LM22
Ordinary epidermal cell wall	+	+++	-	++	++	+++	+	++	+++	+++	-	-
Gland head cell wall	+++	+++	+++	+	+	+	+	++	++	++	-	-
Gland stalk cell wall	++	++	++	+	+	++	++	+	+++	+++	-	-
Ingrowths of gland basal cell wall	+	+++	+++	+	+	+	+	-	+++	+++	-	-

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
