# Peer review of "Immunocytochemical Analysis of the Wall Ingrowths in the Digestive Gland Transfer Cells in Aldrovanda vesiculosa L. (Droseraceae)"

_cells, 2022, doi:10.3390/cells11142218_

Round 1
Reviewer 1 Report
This manuscript reports data on the analysis of cell wall growths in the glands of carnivorous plants. It is certainly an interesting work because there is a lack of information on this topic. The authors analyzed the composition of the cell walls using antibodies directed against different classes of pectins and hemicelluloses, thus highlighting a particular composition of the cell wall ingrowths.
The introduction is well written. The authors mentioned all the most important points, such as the importance of transfer cells and their widespread distribution in the plant kingdom. In the end they clearly explained the scope of the work, which consists in comparing cell wall ingrowths of glandular cells of carnivorous plants with the cell wall ingrowths of other types of transfer cells. No changes are required.
1. In the section of methods, I do not realize what private collection means, from which the authors have taken carnivorous plants. Is it a private botanical garden? Is there no availability of carnivorous plants in public botanical gardens? What about the reliability and reproducibility of the results given that the plants come from private repositories?
2. One point is not clear to me, and that is the stage of development of carnivorous plants. Are the plants the same age? Are the glands taken from plants at the same stage of development? I think more information about the plant source should be provided.
3. The images in figure 1 are informative and appreciable, except for panel F obtained with calcofluor White; the image is of lower quality, moreover the signal is excessive, as if overexposed, so that the cell walls are much thicker than what is perceived by the other images.
4. The signal of AGP obtained with JIM antibodies is important and certainly adequate. In some cases, however, the figure can be improved. For example, immunofluorescences are of sufficient quality, but some images at higher magnification would have helped to see more detail. The magnifications used give the idea of a signal at the level of cell wall ingrowths, but do not allow to deepen. Panel C about Jim8 is not of the same quality as other immunogold images and therefore it would be appropriate to show a better image. In Figure C there is too much background noise, and the gold particles are difficult to observe. What is the strong signal with JIM14 at the bottom of panel J?
5. Why was the immunogold not performed with JIM5? Did the authors rule out any localization of JIM5 epitope in cell wall ingrowths? Other antibodies against pectins were used in immunogold, although the signal had not been observed in cell walls ingrowth. Please add immunogold images with Jim5.
6. The signals of JIM5 and LM19 (two antibodies directed against low-esterified pectins) are not exactly comparable. It would be useful to specify how the two antibodies differ.
7. In general, it would be appropriate to specify more the reason of different antibodies. Why have different antibodies been used to label some specific polysaccharides? What is the different specificity of the antibodies used? This would still help in the discussion.
8. The text lacks the reference to Figure 6 containing the images obtained with the LM21 and LM22 antibodies.
9. Statistical analysis of the immunofluorescence signal can help, but much more information is needed to understand how the data were obtained. Since the various analyses have been carried out on sections that are also quite different, I wonder which sections have been used, which points have been considered. This is because the choice is important in affecting the result. So, it is necessary for the authors to provide much more information on how they measured the intensity of immunofluorescence. The drawings are useful to understand where the data was measured, but it would be important to know how many measurements were taken, at what points, if the different sections have in any way influenced, etc.
In conclusion, the manuscript is interesting. Although it is a report on the presence and relative abundance of several polysaccharide epitopes, this remains the only way to visualize the relative presence and distribution of specific polysaccharides in the cell walls. So, although sometimes this type of work is considered as just a description, for me it has a significant relevance. However, considering the quality of the journal, I ask the authors to consider all the points listed above.
Author Response
Author's Reply to the Review Report (Reviewer 1)
Comments and Suggestions for Authors
This manuscript reports data on the analysis of cell wall growths in the glands of carnivorous plants. It is certainly an interesting work because there is a lack of information on this topic. The authors analyzed the composition of the cell walls using antibodies directed against different classes of pectins and hemicelluloses, thus highlighting a particular composition of the cell wall ingrowths.
We are grateful for all nice words as well as for all suggestions and corrections.
The introduction is well written. The authors mentioned all the most important points, such as the importance of transfer cells and their widespread distribution in the plant kingdom. In the end they clearly explained the scope of the work, which consists in comparing cell wall ingrowths of glandular cells of carnivorous plants with the cell wall ingrowths of other types of transfer cells. No changes are required.
Many thanks for such nice opinion.
- In the section of methods, I do not realize what private collection means, from which the authors have taken carnivorous plants. Is it a private botanical garden? Is there no availability of carnivorous plants in public botanical gardens? What about the reliability and reproducibility of the results given that the plants come from private repositories?
In Poland, we have many enthusiasts (hobbyists) who have wonderful collections of carnivorous plants. Often, these collectors have richer collections of certain plant groups than even botanical gardens. A. vesiculosa is a species that is rarely found in the collections of botanical gardens due to the specific growing requirements. However, it is a species that is not difficult in the taxonomy context and there is no possibility of confusion with other species. In addition, we want to build relationships between scientists and hobbyists to promote science and valuable species.
- One point is not clear to me, and that is the stage of development of carnivorous plants. Are the plants the same age? Are the glands taken from plants at the same stage of development? I think more information about the plant source should be provided.
We added in Material and methods: For the digestive gland analysis mature traps were taken from plants at the same stage of development.
- The images in figure 1 are informative and appreciable, except for panel F obtained with calcofluor White; the image is of lower quality, moreover the signal is excessive, as if overexposed, so that the cell walls are much thicker than what is perceived by the other images.
Thank you for this suggestion, we corrected this image.
- The signal of AGP obtained with JIM antibodies is important and certainly adequate. In some cases, however, the figure can be improved. For example, immunofluorescences are of sufficient quality, but some images at higher magnification would have helped to see more detail. The magnifications used give the idea of a signal at the level of cell wall ingrowths, but do not allow to deepen. Panel C about Jim8 is not of the same quality as other immunogold images and therefore it would be appropriate to show a better image. In Figure C there is too much background noise, and the gold particles are difficult to observe.
Thank you for these suggestion we modified figures.
What is the strong signal with JIM14 at the bottom of panel J?
This signal occurred in wall labyrinth of basal cells.
- Why was the immunogold not performed with JIM5? Did the authors rule out any localization of JIM5 epitope in cell wall ingrowths? Other antibodies against pectins were used in immunogold, although the signal had not been observed in cell walls ingrowth. Please add immunogold images with Jim5.
We performed but we did not show. Now we added.
- The signals of JIM5 and LM19 (two antibodies directed against low-esterified pectins) are not exactly comparable. It would be useful to specify how the two antibodies differ.
JIM5 bind to partially methyl-esterified HG epitope: unesterified residues (up to 40%) adjacent to or flanked by residues with methylester groups, while LM19 binds to low methyl-esterified HG (Vandenbosh et al., 1989).
In many cases LM19 display similar binding properties to JIM5 for e.g. in tobacco stem pith parenchyma (Verhertbruggen et al., 2008) or at early stage of somatic embryogenesis of of Quercus suber (Perez-Perez et al., 2019).
In this study we wanted to see if there any differences in localization of those two PMEs.
Vandenbosch, K. A. et al. Common components of the infection thread matrix and the intercellular space identified by immunocytochemical analysis of pea nodules and uninfected roots. Te EMBO journal 8, 335–341 (1989)
Verhertbruggen Y, Marcus SE, Haeger A, Ordaz-Ortiz JJ, Knox JP. An extended set of monoclonal antibodies to pectic homogalacturonan. Carbohydr Res. 2009 Sep 28;344(14):1858-62. doi: 10.1016/j.carres.2008.11.010. Epub 2008 Nov 27. PMID: 19144326.
Pérez-Pérez Y, Carneros E, Berenguer E, Solís MT, Bárány I, Pintos B, Gómez-Garay A, Risueño MC, Testillano PS. Pectin De-methylesterification and AGP Increase Promote Cell Wall Remodeling and Are Required During Somatic Embryogenesis of Quercus suber. Front Plant Sci. 2019 Jan 8;9:1915. doi: 10.3389/fpls.2018.01915. PMID: 30671070; PMCID: PMC6331538.
- In general, it would be appropriate to specify more the reason of different antibodies. Why have different antibodies been used to label some specific polysaccharides? What is the different specificity of the antibodies used? This would still help in the discussion.
We would like to have more complex view about cell wall polysaccharides in digestive glands, because we would like to continue our study using other carnivorous plant genera (We sent grant application and now we have to wait for result).
- The text lacks the reference to Figure 6 containing the images obtained with the LM21 and LM22 antibodies.
We corrected.
- Statistical analysis of the immunofluorescence signal can help, but much more information is needed to understand how the data were obtained. Since the various analyses have been carried out on sections that are also quite different, I wonder which sections have been used, which points have been considered. This is because the choice is important in affecting the result. So, it is necessary for the authors to provide much more information on how they measured the intensity of immunofluorescence. The drawings are useful to understand where the data was measured, but it would be important to know how many measurements were taken, at what points, if the different sections have in any way influenced, etc.
Statistical analysis was performed on 3 representative glands from 3 different traps (3 measurements). For all measurements of selected types of cell walls ROI (region of interest) was the same - 15 µm2. Measurements were taken from the same magnification (63x) of transverse section of glands.
In conclusion, the manuscript is interesting. Although it is a report on the presence and relative abundance of several polysaccharide epitopes, this remains the only way to visualize the relative presence and distribution of specific polysaccharides in the cell walls. So, although sometimes this type of work is considered as just a description, for me it has a significant relevance. However, considering the quality of the journal, I ask the authors to consider all the points listed above.

Reviewer 2 Report
This is an interesting work, with convincing description and in general good imaging. However, there are several things to do in order to improve the manuscript before acceptance. I have annotated several issues on the attached PDF, with comments and suggestions, that should be addressed. Some of them are also listed here:
1. Please respect the scientific nomenclature. Keep species names in italics everywhere. Also, for brevity, please write A. vesiculosa and D. muscipula, instead of mentioning only the genus.
2. Please describe in detail the protocols for preparation of LR white-embedded samples. The loop of citations among your relevant papers does not help us to understand.
3. There is a problem in some immunogold TEM micrographs: To my experience, contrasting with lead citrate is too heavy, hiding the gold particles. The previous figures, in the paper about D. muscipula, were clear and nice! Please, provide new images with less contrasting.
4. Also omit the pseudo-Schiff propidium iodide procedure, it does not provide any further information.
5. There is a confusion, where you refer to whole traps vs. glands. Also it could help to describe the epidermal cells of the traps as "ordinary epidermal" -which is the correct full name!- to allow discerning from the cells of the glands.
6. The "conclusions" section is just a summary of the observations. Either give real conclusions about the significance of the findings or delete it.
7. You should include a table, to illustrate the presence or absence of each epitope detected in each cell type/wall. This would be very helpful and not time-consuming.
8. One last question of curiosity: How did Dr. MIranda participate to this work? He is not mentioned at all in authors contributions!

Author Response
Author's Reply to the Review Report (Reviewer 2)
Comments and Suggestions for Authors
This is an interesting work, with convincing description and in general good imaging. However, there are several things to do in order to improve the manuscript before acceptance. I have annotated several issues on the attached PDF, with comments and suggestions, that should be addressed. Some of them are also listed here:
We are thankful for all corrections and suggestions. All these are very helpful for us. Corrections are in “red” in the manuscript
- Please respect the scientific nomenclature. Keep species names in italics everywhere. Also, for brevity, please write A. vesiculosa and D. muscipula, instead of mentioning only the genus.
We corrected
- Please describe in detail the protocols for preparation of LR white-embedded samples. The loop of citations among your relevant papers does not help us to understand.
We added detailed protocol.
- There is a problem in some immunogold TEM micrographs: To my experience, contrasting with lead citrate is too heavy, hiding the gold particles. The previous figures, in the paper about D. muscipula, were clear and nice! Please, provide new images with less contrasting.
We agree in some TEM, the gold particles are not so good seen, so we corrected.
- Also omit the pseudo-Schiff propidium iodide procedure, it does not provide any further information.
We agree and this part was deleted.
- There is a confusion, where you refer to whole traps vs. glands. Also it could help to describe the epidermal cells of the traps as "ordinary epidermal" -which is the correct full name!- to allow discerning from the cells of the glands.
Thank you, we included "ordinary epidermal" in the text.
- The "conclusions" section is just a summary of the observations. Either give real conclusions about the significance of the findings or delete it.
We modified:
Because we found that even in the same organ/structure (gland), transfer cells may differ in the composition of the cell wall ingrowths (glandular versus endodermoid cells), further research should be more comprehensive and cover different types of glandular structures and different types of tissue in the analyzed species.
- You should include a table, to illustrate the presence or absence of each epitope detected in each cell type/wall. This would be very helpful and not time-consuming.
We added Table 1.
- One last question of curiosity: How did Dr. MIranda participate to this work? He is not mentioned at all in authors contributions!
We corrected authors contribution part.

Round 2
Reviewer 1 Report
I found that the authors have adequately replied to my previous concerns.
Author Response
Author's Reply to the Review Report (Reviewer 1)
Comments and Suggestions for Authors
I found that the authors have adequately replied to my previous concerns.
Thank you for kind words.

Reviewer 2 Report
I am glad to see that the revised version of manuscript cells-1774065 is indeed significantly improved. However, there are still things to amend before final acceptance.
1. Please include a detailed presentation of the LR White protocol (dehydration, embedding and polymerizing).
2. Still some mistakes in the text have to be corrected.
3. At first mention write full species names, then only abbreviated, eg D. muscipula, throughout the rest of the text.
You will find everything in the annotated PDF, please go through it carefully to fix all issues.

Author Response
Author's Reply to the Review Report (Reviewer 2)
Comments and Suggestions for Authors
I am glad to see that the revised version of manuscript cells-1774065 is indeed significantly improved. However, there are still things to amend before final acceptance.
- Please include a detailed presentation of the LR White protocol (dehydration, embedding and polymerizing).
We included more detailed protocol.
- Still some mistakes in the text have to be corrected.
We corrected in whole text.
- At first mention write full species names, then only abbreviated, eg D. muscipula, throughout the rest of the text.
We corrected.
You will find everything in the annotated PDF, please go through it carefully to fix all issues.
Many thanks for all corrections. We improved also figure. We corrected (changes are in blue).
